# Mitoquinone Helps Combat the Neurological, Cognitive, and Molecular Consequences of Open Head Traumatic Brain Injury at Chronic Time Point

**DOI:** 10.3390/biomedicines10020250

**Published:** 2022-01-24

**Authors:** Muhammad Ali Haidar, Zaynab Shakkour, Chloe Barsa, Maha Tabet, Sarin Mekhjian, Hala Darwish, Mona Goli, Deborah Shear, Jignesh D. Pandya, Yehia Mechref, Riyad El Khoury, Kevin Wang, Firas Kobeissy

**Affiliations:** 1Faculty of Biochemistry and Molecular Genetics, American University of Beirut, Beirut 1107 2020, Lebanon; muhammad.a.haidar@gmail.com (M.A.H.); cab20@mail.aub.edu (C.B.); skm12@mail.aub.edu (S.M.); hd30@aub.edu.lb (H.D.); 2Department of Pathology & Anatomical Sciences, University of Missouri School of Medicine, Columbia, MO 65212, USA; zist7p@missouri.edu; 3Centre de Biologie Integrative (CBI), Molecular, Cellular, and Developmental Biology Department (MCD), University of Toulouse, Centre National de la Recherche Scientifique (CNRS), Université Paul Sabatier (UPS), 31062 Toulouse, France; mahatabet.mt@gmail.com; 4Chemistry and Bioehcmistry Department, Texas Tech University, Lubbock, TX 79409, USA; Mona.Goli@ttu.edu (M.G.); Yehia.Mechref@ttu.edu (Y.M.); 5Brain Trauma Neuroprotection (BTN) Branch, Center for Military Psychiatry and Neuroscience, Walter Reed Army Institute of Research, Silver Spring, MD 20910, USA; deborah.a.shear.civ@mail.mil (D.S.); jignesh.d.pandya.civ@mail.mil (J.D.P.); 6Neuromuscular Diagnostic Laboratory, Department of Pathology and Laboratory Medicine, American University of Beirut Medical Center, Beirut 1107 2020, Lebanon; 7Program for Neurotrauma, Neuroproteomics & Biomarkers Research, Departments of Emergency Medicine, Psychiatry, Neuroscience and Chemistry, University of Florida, Gainesville, FL 32611, USA

**Keywords:** neurotrauma, oxidative stress, neurodegeneration, neuroinflammation, moderate traumatic brain injury

## Abstract

Traumatic brain injury (TBI) is a heterogeneous disease in its origin, neuropathology, and prognosis, with no FDA-approved treatments. The pathology of TBI is complicated and not sufficiently understood, which is the reason why more than 30 clinical trials in the past three decades turned out unsuccessful in phase III. The multifaceted pathophysiology of TBI involves a cascade of metabolic and molecular events including inflammation, oxidative stress, excitotoxicity, and mitochondrial dysfunction. In this study, an open head TBI mouse model, induced by controlled cortical impact (CCI), was used to investigate the chronic protective effects of mitoquinone (MitoQ) administration 30 days post-injury. Neurological functions were assessed with the Garcia neuroscore, pole climbing, grip strength, and adhesive removal tests, whereas cognitive and behavioral functions were assessed using the object recognition, Morris water maze, and forced swim tests. As for molecular effects, immunofluorescence staining was conducted to investigate microgliosis, astrocytosis, neuronal cell count, and axonal integrity. The results show that MitoQ enhanced neurological and cognitive functions 30 days post-injury. MitoQ also decreased the activation of astrocytes and microglia, which was accompanied by improved axonal integrity and neuronal cell count in the cortex. Therefore, we conclude that MitoQ has neuroprotective effects in a moderate open head CCI mouse model by decreasing oxidative stress, neuroinflammation, and axonal injury.

## 1. Introduction

Globally, TBI is one of the leading causes of death and long-lasting disability. The incidence of TBI is estimated to be 939 in 10,000 worldwide, with major causes being falls, vehicle accidents, wars, and sports [1]. The mortality rates worldwide are assessed to be between 7% and 23%, 90% of which are in developing countries [2]. TBI studies in the Middle East are limited; however, it was predicted that the incidence in the region is 45 per 100,000 [2]. Survivors suffering from disabilities endure major socioeconomic burdens as well [3]. TBI leads to neurological complications through two major events, dynamic and overlapping, denoted as primary and secondary injuries. Primary injury defines the severity of TBI and therefore represents the major prognostic factor. Damaged neural cells undergo ionic imbalance characterized by an influx of calcium (Ca^2+^), potassium (K^+^), and glutamate [4]. The resultant excitotoxicity leads to an energetic crisis and oxidative stress by increasing reactive oxygen species (ROS) [5]. These can then directly stimulate the release of cytokines and pro-inflammatory factors, contributing to an elevated inflammatory state [6,7]. Following this, secondary injury is triggered, initiating a cascade of metabolic events that include further excitotoxicity, neuroinflammation, disruption of the blood–brain barrier (BBB), and cell death [8]. Notably, mitochondrial dysfunction, characterized by their membrane disruption and fission/fusion imbalance, plays an important role in the pathology of TBI via the excessive production of ROS, eventually leading to apoptotic cell death [9].

Finding a treatment for TBI remains a major challenge as there are currently no FDA-approved drugs with present approaches mainly targeting the symptoms. Depending on the severity of the injury, these can range from anti-seizure drugs and diuretics to coma-inducing drugs. Anti-seizure prophylaxis is administered to prevent the development of epileptic episodes post-TBI; however, due to the shortage of evidence on its effects in the long term, it is usually discontinued a week after injury [10]. Intracranial pressure is also monitored when individuals experience TBI, and in the case of its elevation, pressure is released by medically inducing venous flow and removing cerebrospinal fluid (CSF) from the intracranial regions [11].

Current animal studies on possible therapies have explored multiple candidates such as calcium channel blockers, erythropoietin, and hypothermia. Calcium channel blockers such as nimodipine and ziconotide have been considered as possible agents to enhance mitochondrial function and improve outcomes in patients with spontaneous subarachnoid hemorrhage [12]. However, the former was later revealed to have no effects on mortality and morbidity in TBI patients, and the latter was shown to lead to numerous side effects such as hypotension [13,14]. On the other hand, other suggested treatments target oxidative stress since antioxidants have been shown to ameliorate the pathology of TBI [15,16]. An example of these is the mitochondria-targeted drug mitoquinone (MitoQ). It is synthesized by conjugating a ubiquinone moiety to a triphenylphosphonium cation (TPP^+^) [17]. TPP^+^ is a lipophilic cation that directs the ubiquinone moiety to the inner mitochondrial membrane (IMM) and enables its accumulation due to the high electrochemical potential [18]. Accumulated MitoQ in the IMM is reduced by complex II into its antioxidative quinol form [19], noting that almost all molecules that enter the mitochondria are adsorbed to the matrix surface of the IMM where they are constantly recycled [20]. MitoQ can easily cross the BBB [21] and has been shown to act by activating the nuclear factor erythroid 2 (NFE2)-related factor 2 (Nrf2) pathway (Nrf2-ARE) [22]. Previous studies from our laboratory (currently under review) have shown that MitoQ improved the outcomes of closed head repetitive mild TBI in mice at acute, subacute, and chronic time points. Our current study aimed at assessing MitoQ’s neuroprotective effects on the long-term behavioral, cognitive, and molecular outcomes in a moderate open head TBI mouse model. We hypothesized that improved outcomes are carried out by MitoQ’s ability to dampen the state of oxidative stress, decrease cellular injury, and ameliorate neuroinflammation. We were able to show that MitoQ indeed exhibits neuroprotective properties in a moderate controlled cortical impact (CCI) mouse model, with effects translating to the molecular and behavioral/cognitive levels. 

## 2. Materials and Methods

### 2.1. Animals

Thirty-four C57BL/6 male mice were housed in a controlled environment (12 h light/dark cycles, 22 ± 2 °C). The study was carried out at the Animal Care Facility of the American University of Beirut (AUB) and all animal experiments were performed in compliance with the AUB Institutional Animal Care and Use Committee (IACUC) guidelines with the reference number 17-01-458, 1 February 2019. All animals were handled under pathogen-free conditions and fed chow diet ad libitum. 7–8 weeks old male mice were divided into three groups: Sham, CCI, and CCI + MitoQ. The mice in the CCI and CCI + MitoQ groups were subjected to TBI using a controlled cortical impact machine (CCI). Thirty minutes after TBI, mice in the CCI + MitoQ group were injected intraperitoneally (IP) with MitoQ at a dosage of 8 mg/kg administered three times per week over 30 days, starting at 30 min post-injury. The dose was adopted from a previous study on MitoQ and TBI [22].

Keeping in mind that the test most prone to variability is the Morris water maze, we used it to determine the number of mice required for cognitive and behavioral tests. To achieve 90% power to detect a difference with 95% confidence, 9 animals were required. Therefore, we assigned 11–12 mice per group for behavioral, cognitive, and neurological tests: group 1: Sham (*n* = 10); group 2: CCI (*n* = 10); and group 3: CCI + MitoQ (*n* = 10). Animals were assigned to experimental groups by randomly selecting which animals received the treatment from those that underwent the injury. Animals that showed weak health post-surgery and exhibited any disease condition were not included in the study as per the IACUC regulations. Following sacrifice, four animals per group (*n* = 4) were used to perform immunofluorescence and another four animals (*n* = 4) per group for the RT-qPCR.

### 2.2. Stereotaxic Surgery: Controlled Cortical Impact

Each mouse was anesthetized with a ketamine/xylazine mixture (50 mg/kg and 15 mg/kg, respectively) administered intraperitoneally. Each mouse was then fixed on a stereotaxic frame, and a longitudinal skin incision was made in the middle of the mouse’s head by a surgical scalpel to expose the skull. An ointment (Xailin^®,^, Nicox, France) was applied to the eyes to protect vision during surgery. Using a drill and forceps, the part of the skull above the somatosensory area of the parietal lobe was removed to expose the brain. The injury was created using the Leica Impact One Angle Controlled Cortical Impact (CCI) machine. The center of the impactor was placed above the somatosensory area of the parietal cortex of the brain using the following coordinates: +1.0 mm AP, +1.5 mm ML, and −2 mm DV. The duration of impact was kept constant with a dwell time of 1 s, at a velocity of 4 m/s. The depth of the injury was set to 1.5 mm. The tip of the impactor was 1 mm in diameter. The size of the bone flap was 1.7 mm in diameter, which was carefully removed using a manual trephine to expose the dura matter without damaging it. The wound was closed using silk sutures (MERSILK™-W587H, Ethicon, OH, USA). Each mouse was placed on a heating pad to maintain the body’s temperature. The CCI + MitoQ group received the first MitoQ injection 30 min following injury. For the animals in the Sham group, drilling was performed to remove confounding factors.

### 2.3. Mitoquinone Supplementation

Mitoquinone (Focus Biomolecules, 10-1363, MW = 663.64) was prepared by dissolving 25 mg in 10% dimethylsulfoxide (DMSO) (100 μL in 900 μL phosphate-buffered saline (PBS)). Further dilutions using PBS were carried out to obtain working solutions of 1 mg/mL. MitoQ was intraperitoneally (IP) supplemented at a dose of 8 mg/kg three times per week over 30 days, starting at 30 min post-injury. The experimental timeline is presented below in Figure 1A.

### 2.4. Garcia Neuroscore

The neuroscore was adopted from Garcia et al. to assess the integrity of neurological function via different criteria [24]. Every group was tested 3 days before CCI and then on days 3, 7, and 30 after the injury. The animals underwent six different evaluations on every test day, each being scored from 0 to 3, according to Table 1.

### 2.5. Grip Strength Test

This grip strength test was performed to assess endurance of motor skills and muscular dysfunction using the 4700-grip strength meter (UGO BASILE^®^, Gemonio, Italy). Each animal was held by its tail and was allowed to catch a trapeze-shaped metal object with both of its paws. A total of three trials were conducted for each animal. Muscle strength in gram force (gf) was recorded along with the total time of grip and the time the mouse applied the force. For analysis, the average of muscle strength, in gf, in the three trials, was normalized to the mouse weight [25].

### 2.6. Pole Climbing Test

The pole climbing test was conducted to evaluate motor coordination. The apparatus consisted of a metal pole with a length of 60 cm and a diameter of 1 cm, set perpendicularly in a big cage. The pole was wrapped with tape to facilitate the animal’s grip on the apparatus. Animals were habituated to descend on the pole for three trials before the recorded testing. The mice were placed on top of the pole with their heads directed upwards and allowed to descend freely [26]. The time needed for each mouse to reach the bottom of the pole (total time) was recorded. For each animal, a total of three testing trials were conducted on the testing day.

### 2.7. Adhesive Removal Test

This test was performed by placing a rectangular adhesive tape strip on each mouse’s nostril to assess sensorimotor function. Then, the animal was placed back in its cage. Two different values were recorded using two stopwatches: (i) the time needed to make the first contact with the tape, which represents nose sensitivity, and (ii) the time needed to remove the tape, which represents dexterity.

### 2.8. Forced Swim Test

This test was used to assess despair and depression-like behavior. Animals were placed in a glass cylinder (20 (height) × 11 cm (diameter)) filled with 25 °C water to a depth of 15 cm for 6 min. The 6 min included a 6 min acclimation period at the beginning of the test, followed immediately by 5 min of testing. Depression was reflected by the time mice spent floating on the surface of the water without moving. Animals were then dried with a towel and returned to their heated cage. The behavior of the mice was recorded using a camera, and immobility time was analyzed later during the last five-minute period of the test. Immobility time was defined as the time a mouse spent floating and only making necessary movements to keep its body balanced and head above the water.

### 2.9. Morris Water Maze Test

The Morris water maze (MWM) was used to study spatial learning and memory. The apparatus consisted of a circular pool (110 cm diameter, 55 cm depth) half-filled with water and maintained at 22 °C. A 10 cm-diameter platform was placed in the water. Three extra-maze cues were located around the pool to spatially guide the mice. Black and white cues were used to omit any variance in color discrimination among mice. Non-toxic white paint was added to the water surface to make it opaque and ensure accurate tracking. The ANY-maze 5.2 software (Stoelting Co., Wood Dale, IL, USA) was used to track the movement of mice and record the different parameters to be evaluated during the five days of testing. 

On the first day, the platform was made visible by mounting a flag for cued trials. Four trials were carried out for each animal where the position of the platform and the starting position of animals changed among trials. On days two through four, which are the acquisition days, the flag was removed, and the platform was fixed in the northeast (NE) quadrant submerged in the water. However, the starting position of the animals varied among trials where three trials were carried out. The maximum time of a trial was set to one minute. If the animal failed to find the platform during this time, it was guided to the platform by the experimenter, where the animals were allowed to sit on the platform for 20 s for memory consolidation. If the animals found the platform before this time, the test was considered completed, and the animals were kept on the platform for five seconds. On day five, or the probe trial day, the platform was removed, and the mice were allowed to swim for 1 min for one trial only. After each trial on all days, the mice were dried with a towel and allowed to rest in a heated cage.

### 2.10. Novel Object Recognition Test

Novel object recognition depends on the innate capacity of rodents to discriminate a novel object from a familiar object (previously encountered). Mice were placed in the center of an open field at the beginning of each trial and freely explored the open field and objects. At the end of each trial (5 min), mice were removed from the open fields and placed in their home cages next to each testing area for the inter-trial interval (ITI—5 min). The open fields and objects were cleaned with 70% ethanol during the ITI. During Trial 1 (learning trial), the animals explored an open field with two identical objects. During trial 2 (testing trial), one object was kept in the testing field, while the other was replaced by a novel object for exploration. All data were recorded by a video camera that was connected to image analyzer software Any-Maze 5.2 (Stoelting Co., Wood Dale, IL, USA). The zones were located 5 cm around each object on the software, and the software recorded the time the animal spent in each zone.

### 2.11. RT-qPCR

RNA was extracted using Trizol (T9424-100ML, Sigma Aldrich, St. Louis, MO, USA), and then any contaminating genomic DNA was removed using TURBO DNA-free™ Kit (AM1907, Thermo Fisher Scientific, Waltham, MA, USA) according to the manufacturer’s instructions. An amount of 1.5 μg of total RNA was reverse transcribed using the iScript™ cDNA Synthesis Kit (1708890, Bio-Rad, Hercules, CA, USA). Quantitative real-time PCR was applied to 1 μL of the obtained cDNA using the Quantifast^®^ SYBR^®^ Green PCR Master Mix (204054, Qiagen, Hilden, Germany) with 10 μM of each of the reverse and forward primers. Cycling conditions were as follows: 95 °C for 10 min for one cycle, then 95 °C for 10 s followed by 60 °C for 30 s and 72 °C for 30 s for 40 cycles, and, finally, 72 °C for 5 min for one cycle. The primers used are listed in Table 2.

### 2.12. Immunofluorescence Staining

The animals allocated for immunofluorescent staining underwent perfusion right after terminal tissue collection. Prior to surgery, the mice were anesthetized via IP injection of ketamine and xylazine solution (50 mg/kg and 15 mg/kg, respectively). A lateral incision was made through the integument and abdominal wall beneath the rib cage. The diaphragm was then incised, and the pleural cavity was exposed. The sternum was then lifted and any tissue connecting it to the heart was cut. A perfusion needle was introduced into the left ventricle without reaching the aorta, and an incision was made to the animal’s right atrium. PBS was firstly perfused into the animals until the fluids were running clear and the liver was clear as well. When this occurred, PBS was switched with PFA until fixation tremors were observed. Brain tissue was placed in PFA for 48 h before being transferred into 30% sucrose solution for further preservation.

In preparation for immunofluorescence staining, brain tissue underwent sectioning using a microtome and was stored as free-floating sections in sodium azide. Free-floating brain sections (40 μm thick) were washed with PBS followed by PBST (0.1% Triton in PBS) and then incubated for 1.5 h in a blocking solution of 10% heat-inactivated fetal bovine serum (FBS) in PBST. Tissues were then incubated overnight at 4 °C with primary antibodies, diluted in 1% FBS solution. The primary antibodies used were anti-GFAP (1:1000 dilution; MCA-5C10, Encor Biotechnology, Gainesville, FL, USA) as an astrocyte marker; anti-NeuN (1:1000 dilution; RPCA-FOX3-AP, Encor Biotechnology) as a marker of mature neurons; anti-Iba-1 (1:1000 dilution; 019-19471, Fujifilm Wako Chemicals, Richmond, WV, USA) as a microglia marker; anti-MBP (1:1000 dilution; MCA-7D2, Encor Biotechnology) as an axonal marker. Then, sections were rinsed in PBST and incubated with the appropriate fluorochrome-conjugated secondary antibody (1:1000 dilution) for 1 h at room temperature, followed by three washes in PBST. The secondaries used were Donkey Anti-Mouse IgG H&L Alexa Fluor^®^ 488 (ab150105, Abcam, MA, USA) and Goat Anti-Rabbit IgG H&L (Alexa Fluor^®^ 568) (ab175696, Abcam, MA, USA). Finally, all sections were counterstained in 1 μg/mL of Hoechst (Sigma Aldrich), diluted in PBS, and mounted using Fluoromount (F4680-25ML, Sigma Aldrich). Microscopic imaging was conducted using a Zeiss LSM 710 confocal microscope. Images were acquired as tile scans with 40× oil objectives and analyzed using the Zeiss ZEN 2009 image analysis software and NIH ImageJ program. Images for the different experimental interventions were acquired under the same laser and microscopic parameters for the purpose of consistency. At least three sections per mouse were used for quantification with different levels to cover the hippocampus and cortex areas. For the hippocampus, quantification was conducted along the CA3 and the dentate gyrus (DG). For the cortex, the entire area around the injury in the somatosensory cortex was captured via tile scanning (Figure 1B). The fields from all three sections were averaged for every animal.

### 2.13. Statistical Analysis

Statistical analysis was conducted using GraphPad Prism 9.0 (La Jolla, CA, USA). The Kruskal–Wallis test was performed to compare the datasets that contained more than two groups. Then, Dunn’s test was used to compare the three groups simultaneously. All results were considered significant at a *p*-value of < 0.05: *** (*p* < 0.001), ** (*p* < 0.01), * (*p* < 0.05).

## 3. Results

### 3.1. MitoQ Improves General Neurological Function by Ameliorating Motor Coordination, Muscle Strength, and Sensorimotor Function

We investigated the general neurological function of TBI mice using the modified Garcia neurological score test. The experiment was first conducted 3 days before the injury to make sure that all animals (*n* = 10 per group) had a similar basal score and no prior neurological dysfunction was present (Figure 2A). As expected, all animals had a perfect score, showing that they were eligible to be included in the study. The test was then repeated at 3, 7, and 30 days post-injury (DPI). On day 3 post-injury, the difference between the CCI + MitoQ and CCI groups was not significant. However, at 7 and 30 days post-injury, the CCI + MitoQ group performed significantly better compared to the CCI group (*p* < 0.05). This first step showed us that MitoQ improved overall neurological performance in mice. However, to be able to further assess this, additional testing was performed, from 21 to 23 DPI, to evaluate specific sensorimotor deficits and observe overall motor coordination.

To start with, the adhesive removal test was performed to assess sensorimotor function by observing sensitivity and dexterity. The CCI + MitoQ group performed significantly better than the CCI group by establishing first contact with the adhesive faster (*p* < 0.05) (Figure 2B,C). This showed improved sensitivity. Additionally, the CCI + MitoQ group required less time to remove the adhesive (*p* < 0.05), indicating better dexterity.

Then, motor function was investigated by exploring motor coordination via the pole climbing test, and muscle strength using the grip strength test. The CCI + MitoQ group was able to perform better by taking less time to descend the pole (*p* < 0.05) (Figure 2E) and revealed increased muscle strength (*p* < 0.001) (Figure 2D) compared to the CCI group. Therefore, MitoQ administration helped improve gross motor function. It is noteworthy here that the data show no significant difference between the CCI and Sham groups in the pole climbing test due to increased variability. 

### 3.2. Learning Deficits and Recognition Memory Dysfunction Are Alleviated by MitoQ 

MitoQ’s ability to potentially alleviate cognitive deficits that result from CCI was subsequently evaluated using the Morris water maze (MWM) test. No differences in latency to the platform were found among groups on day 1 of the test, which was performed 25 DPI. During the acquisition days of the test, CCI mice took more time to reach the hidden platform as compared to the Sham and CCI + MitoQ groups. This delay was significant on days 2 (*p* < 0.05), 3 (*p* < 0.001), and 4 (*p* < 0.05) of the acquisition days (Figure 3A). The data indicate that CCI results in a cognitive learning deficit reversed upon MitoQ administration. On the probe trial day of the test, there were differences in the latency to reach the target quadrant by the CCI group, indicating that retention deficits are caused by CCI in the model (Figure 3B,C). However, this was shown to be attenuated by MitoQ since the CCI + MitoQ group required less time to reach the target quadrant (*p* < 0.05) and spent more time in it (*p* < 0.05) (Figure 3B,C).

The novel object recognition (OR) test was conducted for further learning assessment and recognition memory testing. On the testing day, after carrying out habituation and training for the mice in the platform, the total exploration time for all three groups showed no significant difference. This ensured that all animals had an equal chance of exploring the platform and thus ensured the viability of the test (Figure 3F). During the testing trials, the CCI group exhibited no differentiation between familiar and novel objects (*p* < 0.01). This was not the case in the CCI + MitoQ group, which spent more time exploring the novel object, resulting in a positive differentiation index (*p* < 0.05) (Figure 3G).

### 3.3. MitoQ Decreases Depressive-like Behavior

To understand the effects of MitoQ on depressive-like behavior, the forced swim test allowed quantifying hopelessness in all study groups by measuring the immobility time. The CCI-only group showed an increased immobility time when compared to Sham (*p* < 0.001). Of interest, the CCI + MitoQ group demonstrated a decrease in the immobility time compared to the CCI group (*p* < 0.05) (Figure 4). Therefore, MitoQ administration helped in reducing depressive-like behavior in mice.

### 3.4. MitoQ Upregulates Antioxidative Enzymes

The expression level of Nrf2 and downstream antioxidant enzymes, particularly CAT and SOD2, was quantified by RT-qPCR to draw similarities between previous studies and make sure MitoQ acted on the same axis in this experiment. After collecting ipsilateral cortices 30 DPI, the experiments were performed on total mRNA obtained from all three groups (*n* = 4). For all three genes of interest, MitoQ treatment resulted in a significant improvement in expression as compared to CCI (*p* < 0.05) (Figure 5). This means that MitoQ increased the expression of antioxidative enzymes via the Nrf2–ARE axis. 

### 3.5. MitoQ Diminishes Chronic Activation of CNS Inflammatory Cells, Neuronal Death, and Axonal Damage

We attempted to assess if the decreased state of oxidative stress could be translated into ameliorated cellular recovery post-CCI. For this reason, immunofluorescence staining was performed to examine the activation of glial cells, axonal damage, and neuronal loss (*n* = 4 for every group, three fields were quantified for every *n* and averaged). Ionized calcium-binding adaptor protein-1 (Iba-1), an actin-binding protein, was used to stain activated microglia using immunofluorescence. The results show that MitoQ administration significantly attenuated the number of activated microglia both in the ipsilateral cortex (*p* < 0.05) and hippocampus (*p* < 0.05) compared to that of the CCI group (Figure 6A,B). Quantifying the number of microglia in both brain regions was not enough without examining the morphology of these cells. The CCI group exhibited amoeboid and elongated microglia with thick extensions, representing an activated phenotype [27] (Figure 6C). This, however, was not the case in the Sham and CCI + MitoQ groups, where microglia retained their ramified shape with extended ramification, indicative of the resting surveillant phenotype. Taken together, these data show that MitoQ reduces microgliosis following CCI, which is indicative of decreased chronic inflammation. Further investigation was conducted by assessing astrogliosis using the astrocytic activation marker GFAP.

Glial fibrillary acidic protein (GFAP) is an intermediate filament protein used as a marker of reactive astrocytes. The intensity of its expression was used as a marker of active astrocytes. GFAP expression was significantly higher in the ipsilateral hippocampus and cortex of the CCI group compared to that of the Sham group (*p* < 0.05). It was noticeable that activated astrocytes congregated around the injury site and were absent in the contralateral side of the cortex of the injured group (Figure 7A). Such a result conforms with the focal nature of the injury. Upon quantification, our data show that this state of increased GFAP expression was significantly improved by MitoQ administration in the cortex (*p* < 0.05) and the hippocampus (*p* < 0.05) (Figure 7B,C). Likewise, the morphology of activated astrocytes was noticeable through Z-stack imaging of the dentate gyrus (DG), where they exhibited abundant arborization exclusively in the CCI group and retained an inactive morphology in the CCI + MitoQ group (Figure 7D).

Next, we investigated the neuronal cell count to assess cell death. Immunofluorescence staining for NeuN, a nuclear protein found in the nuclei and perinuclear cytoplasm of neurons, was used to assess the number of mature neurons in a semi-quantitative manner [28]. Our results show a significant decrease in the neuronal cell count in CCI as compared to Sham in the cortex and the hippocampus (*p* < 0.05) (Figure 8), with significant improvement in the cortex (*p* < 0.05), but not in the hippocampus, in the CCI + MitoQ group. 

To measure axonal integrity, which has been shown to correlate with cognitive and behavioral deficits [29], we investigated myelin basic protein (MBP). Used as a marker for myelination, MBP is responsible for the adhesion of myelin to the cytosolic surfaces of cells [30]. Prominent axonal damage was observed in the cortex of the CCI group, with characteristic axonal shearing and blebbing (Figure 9B). However, in the CCI + MitoQ group, the axons exhibited a more refined integrity similar to that shown in the Sham group. Quantification of the MBP intensity in the cortex showed a significant decrease in the CCI group compared to the Sham group. This was significantly increased upon MitoQ administration (Figure 9D,E). Unlike the cortex, the hippocampus did not exhibit any differences in the integrity of MBP between all three experimental groups, retaining the focal nature of the injury.

## 4. Discussion

TBI involves a plethora of pathological pathways that contribute to its heterogenic nature, making it extremely difficult to understand and treat. This explains the lack of FDA-approved drugs against TBI, exacerbating the need to investigate potential therapeutic approaches. For that purpose, this study was conducted to assess the potential protective effects of MitoQ in a CCI mouse model. Previously, MitoQ was demonstrated to ameliorate pathological effects of TBI by decreasing neuronal apoptosis, improving neurobehavior, and enhancing antioxidative capacity in the cortices of mice subjected to mild TBI (mTBI) [22]. Other studies exploring the neuroprotective properties of MitoQ in animal models of Huntington’s disease, Alzheimer’s disease (AD), and anterolateral sclerosis (ALS) also showed it can improve memory, enhance motor function, and decrease amyloid-beta (Aβ) accumulation [31,32,33]. In compliance with this, results from our laboratory further support the claim that MitoQ can decrease neurological and cognitive deficits in mTBI at a chronic time point (paper in publication).

In our study, MitoQ was supplemented at a concentration of 8 mg/kg, adapted from Zhou et al., who investigated the effects of MitoQ in experimental traumatic brain injury for the first time in vivo [22]. During the study, different doses of MitoQ were administered to compare dose-dependent effects. The injured mice that received MitoQ at 8 mg/kg exhibited improved neurological severity scores compared to the vehicle-treated group, as well as decreased edema. There was a lack of any improvement in both parameters when the mice received a dose of 2 mg/kg. This study showed that administering MitoQ at 8 mg/kg was safe and revealed neuroprotective effects in a mouse model of weight drop TBI. Similarly, in another study by Xiao et al., MitoQ was administered in mice at a concentration of 5 mg/kg IP to investigate its effects through the Nrf2 pathway [34]. This concentration yielded significant results, where MitoQ was able to provide protective effects in tubular injury via mitophagy.

Any concerns on the side effects of MitoQ, especially its administration for long periods (30 days in our case), have also been addressed in previous studies. There is a comprehensive body of literature showing this in both clinical and pre-clinical trials. Gane et al. showed that MitoQ helped decrease liver damage in hepatitis C patients with no adverse side effects over 28 days [21]. More recently, Rossmann et al. studied the effect of MitoQ on vascular function in adults. The study was conducted over six weeks and showed no significant adverse side effects following administration [35]. Similarly, Rodriguez-Cuenca et al. showed that long-term administration of MitoQ on wild-type mice for up to 28 weeks was safe and did not act as a pro-oxidant or cause damage over this period [36].

Having said that, the first step in our study was to observe the effects of MitoQ on a macroscale and thus investigate its potential in improving cognitive and neurological functions post-CCI. To start with, general neurological testing was carried out via the modified Garcia neurological score test. This allowed scoring the animals on a scale from 0 to 18 to assess general sensorimotor function, motor skill, and coordination-based functions [24]. In this study, MitoQ was shown to improve overall neurological integrity by resulting in better scores at 7 and 30 DPI, which represented the subacute and chronic phases of injury. With these results, we decided to investigate which aspects of the neurological functions were affected by CCI and which were improved by MitoQ.

The adhesive removal test showed that the current CCI model induced sensorimotor deficits in mice, where they took longer to establish contact with and remove tape that was placed on their noses. However, MitoQ was able to shorten this time, showing that it improved sensorimotor function at a chronic time point. To be more specific, this test provided us with the insight that MitoQ improved both dexterity and sensitivity. On the other hand, during the pole climbing test and grip strength test, the current CCI model displayed decreased muscle strength and adversely affected motor coordination, both of which were improved by MitoQ. Although there was a significant difference between the CCI and CCI + MitoQ groups in the pole climbing test, there was no significance between the CCI and Sham groups. This could either be due to high variability in the test scores or the injury model not being sufficient to induce motor coordination dysfunction. In either case, further investigation is warranted. In addition to this, MitoQ demonstrated a positive impact on depressive-like behavior during the forced swim test, revealing that it helped with the notion of hopelessness. This is especially important as TBI has been linked to many neuropsychiatric sequelae in the long run [37,38].

Moreover, CCI resulted in learning deficits and impairment of recognition memory, as seen in the MWM and OR tests. Firstly, during the former, the CCI group displayed latency to reach the platform on the learning days and a tendency towards worsened memory retention, as seen by the latency to reach the target quadrant and the time spent in it on the probe trial day. During the latter, the CCI group showed no differentiation between the familiar and novel objects, revealing recognition memory dysfunction. These findings align with other studies describing increased latency to reach the hidden platform in mice subjected to TBI [39,40,41]. However, when animals received MitoQ for 30 days, they exhibited improved learning and memory retention. Their ability to recognize and differentiate novel from familiar objects was also significantly improved, showing enhanced recognition memory. Therefore, we were able to demonstrate two things in this part of the study: (a) the animal model used was sufficient to induce cognitive, neurological, and behavioral dysfunctions, in compliance with the previous literature [42,43,44,45]; and (b) MitoQ administration showed exciting potential in improving cognitive dysfunctions in the mice at a chronic time point.

Having seen these data on the macroscale, it was important to assess how MitoQ could affect different pathways involved in TBI pathology at a molecular level. We needed to investigate this in the cortex since it retained the injury, and the hippocampus since it is involved in emotions, memory, and other functions impaired in TBI and the CCI model here [46,47,48]. The observed alterations in cognitive functions translate to damaged neuronal structures in the cortex and hippocampus [44,49]. Due to synapse disruption and other pathological effects such as chronic inflammation, oxidative stress, and glutamate altercations, circuit alterations have been linked to memory impairment in TBI and other neurological deficits [50,51,52]. We were able to see in the previous data that MitoQ treatment following CCI resulted in enhanced learning ability and recognition memory, pointing at its potential effects on oxidative stress, chronic inflammation, and neuronal circuitry disruption. This is especially true when referring to a previous study by Zhou et al., which showed MitoQ’s potential in decreasing neuronal apoptosis and enhancing Nrf2 nuclear localization to induce antioxidative effects [22]. For this reason, we performed RT-qPCR analysis to observe how effective MitoQ was, in the current animal model, in acting on the previously described Nrf2-ARE pathway [22]. In our experiment, MitoQ significantly increased the expression of the Nrf2 transcription factor and its downstream antioxidative enzymes SOD2 and CAT. This showed us that it worked on the antioxidative pathway involving Nrf2 and thus exerted its antioxidative effect as expected. After examining the effect of MitoQ on antioxidative enzymes, it was important to investigate other parameters that might be related to improved cognition and neurological function. This was conducted via immunofluorescence.

Previous studies have shown that TBI was followed by microglia and astrocyte activation, leading to several neurological and cognitive dysfunctions [53,54,55,56,57]. This is accompanied by neuronal cell loss via activation of the cell death machinery [58,59,60]. The described CCI model exhibited results that are in line with these previous studies. There was an increased expression of GFAP and Iba-1, showing the presence of chronic activation of astrocytes and microglia, indicative of an active inflammatory state [54,57]. The injury also led to a decreased neuronal count, as examined using the neuronal marker NeuN. This pathology in the cortex and the hippocampus is reflected in the depressive-like state exhibited by the mice, and by the disrupted neurological function and cognition. It must also be highlighted that prominent axonal damage and shearing were observed in the cortex with decreased MBP reactivity due to the breakdown of the protein, indicating disrupted circuitry. However, MitoQ was able to help decrease these effects, where astrocytes and microglia showed dampened activation in the cortico-hippocampal system, and there was reduced neuronal cell loss in the cortex. This decreased activation was apparent in the morphology of both glial cells in the ipsilateral cortices and hippocampi of the mice that received the drug. The used CCI model induced a prominent presence of activated intermediate microglia denoted as bushy microglia, characteristic of their activated morphology [27].

The mitigated neuroinflammatory profile may be related to the antioxidant activity of MitoQ that enhances oxidative enzyme production and therefore prevents excessive production of ROS. Enhanced ROS production is known to directly stimulate the expression of the pro-inflammatory interleukins IL-1α and IL-1β within hours after injury [6,7]. Secreted pro-inflammatory cytokines subsequently contribute to the activation of Ca^2+^-dependent proteolytic enzymes that activate the cell death machinery and induce a pro-inflammatory state [61]. However, these protective effects could also pertain to unexplored pathways that require further investigation. Therefore, this cycle of oxidative stress, neuroinflammation, and neuronal cell death holds potential for many questions that need to be addressed in MitoQ administration. The proposed mechanisms of MitoQ’s effect are summarized in Figure 10.

There are a number of limitations that should be addressed. In this work, only male mice were used, which ignores the female hormonal system that has been shown to have effects in TBI [62]. Additionally, between the animal groups, there was no vehicle-only group because MitoQ was injected in PBS, and previous data from our laboratory showed no difference between the vehicle and Sham groups.

## 5. Conclusions

This study shows that MitoQ administration appears to efficiently reduce the deleterious consequences associated with moderate open head TBI at both the molecular and behavioral/cognitive levels. This paves the way towards identifying a potential therapy for TBI that may improve the quality of life for those affected by it. Moreover, the obtained data provide an additional line of evidence on the potential role of oxidative stress and chronic inflammation in the pathology of brain disorders by mechanisms that need to be unraveled. This also opens more questions on the effects of this supplement on the mitochondrial complexes and neuronal regeneration. This emphasizes two important next steps that need to be addressed. The first is exploring further pathways implicated in the activity of MitoQ in moderate TBI. The second is conducting further assessment of biomarkers of neurodegenerative disorders to check the extent of MitoQ’s protective effects at chronic time points.

## Figures and Tables

**Figure 1 biomedicines-10-00250-f001:**
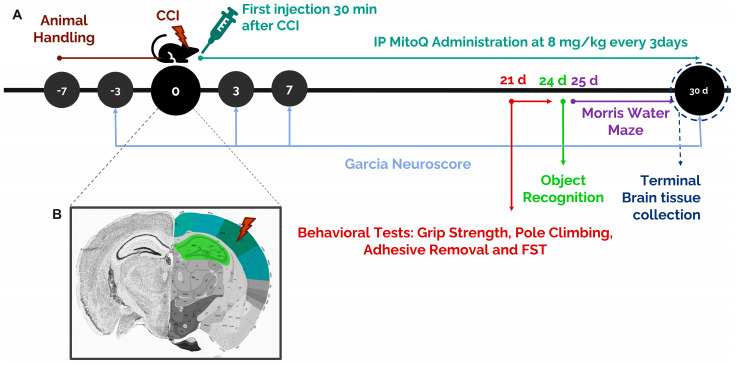
Experimental timeline of the study. (**A**) Mice were divided across three groups: Sham, CCI, and CCI + MitoQ. MitoQ injections were supplemented starting 30 min after the first injury via IP injections at a dose of 8 mg/kg every 3 days for 30 days. Behavioral and cognitive testing commenced after day 21 of the experimental timeline and continued until day 30. (**B**) A coronal section showing the site of injury and the brain regions (green and blue) that were investigated for quantification parameters. The section was obtained from the Allen Mouse Brain Atlas and Allen Reference Atlas—Mouse Brain [23].

**Figure 2 biomedicines-10-00250-f002:**
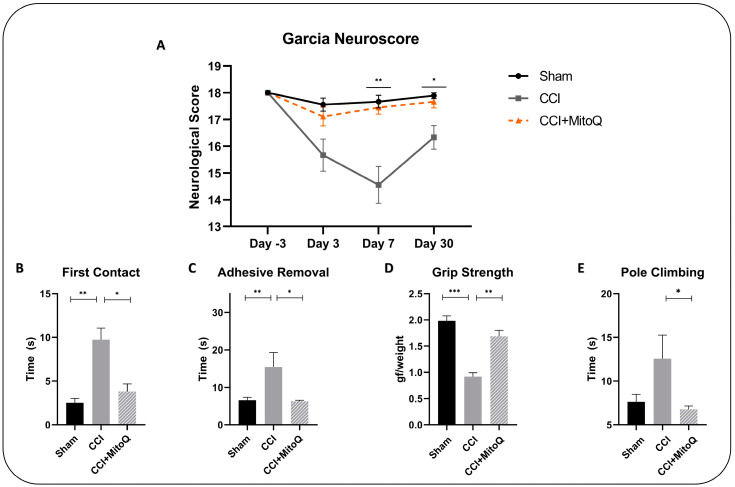
MitoQ ameliorates neurological deficits post-CCI. (**A**) Garcia neuroscore allowed testing for neurological function in mice and showed that MitoQ improved overall neurological performance 7 and 30 days post-CCI. Further testing was conducted to assess sensorimotor function via the adhesive removal test. Mice that received MitoQ took less time to establish contact (**B**) and were able to remove the adhesive faster (**C**) than the CCI group. Additionally, gross motor function was investigated by two tests. (**D**) The grip strength test showed that MitoQ improved overall muscle strength, and the pole climbing test (**E**) showed that CCI + MitoQ had better motor coordination. All tests were executed on Sham (*n* = 10), CCI (*n* = 10), and CCI + MitoQ (*n* = 10). * (*p* < 0.05); ** (*p* < 0.01); *** (*p* < 0.001).

**Figure 3 biomedicines-10-00250-f003:**
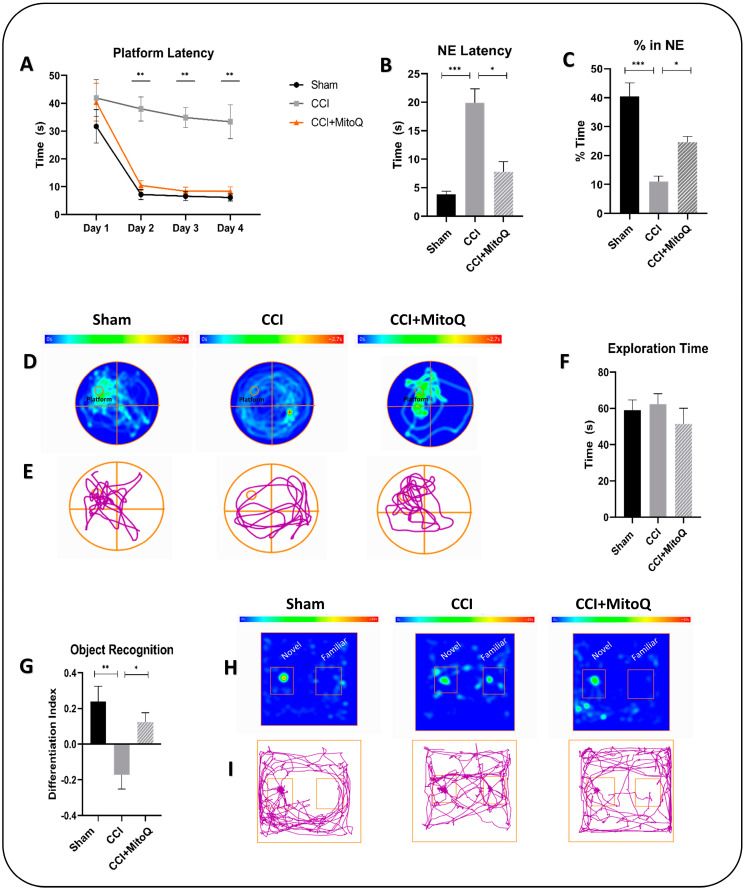
MitoQ improves special learning and recognition memory. Spatial learning and memory in mice from the three groups were evaluated using the Morris water maze. (**A**) shows the time taken for the first arrival at the platform by mice from the three study groups on the acquisition days where the mice learned how to reach the platform based on visual cues. (**B**) shows the time to reach the target quadrant (NE) on the probe trial day. (**C**) shows the percentage of time spent in the target quadrant. (**D**,**E**) show the representative heat maps and trace plots on the probe trial day, respectively. (**H**,**I**) show the representative heat maps and trace plots on the testing day of the object recognition test. For further investigation of memory acquisition, the novel object recognition test was performed. (**F**) Exploration time for all three groups was recorded, and (**G**) the discrimination index was calculated, showing that recognition memory was enhanced in the CCI + MitoQ group. Data represent the mean ± SEM (*n* = 10 per group). * (*p* < 0.05), ** (*p* < 0.01), *** (*p* < 0.001).

**Figure 4 biomedicines-10-00250-f004:**
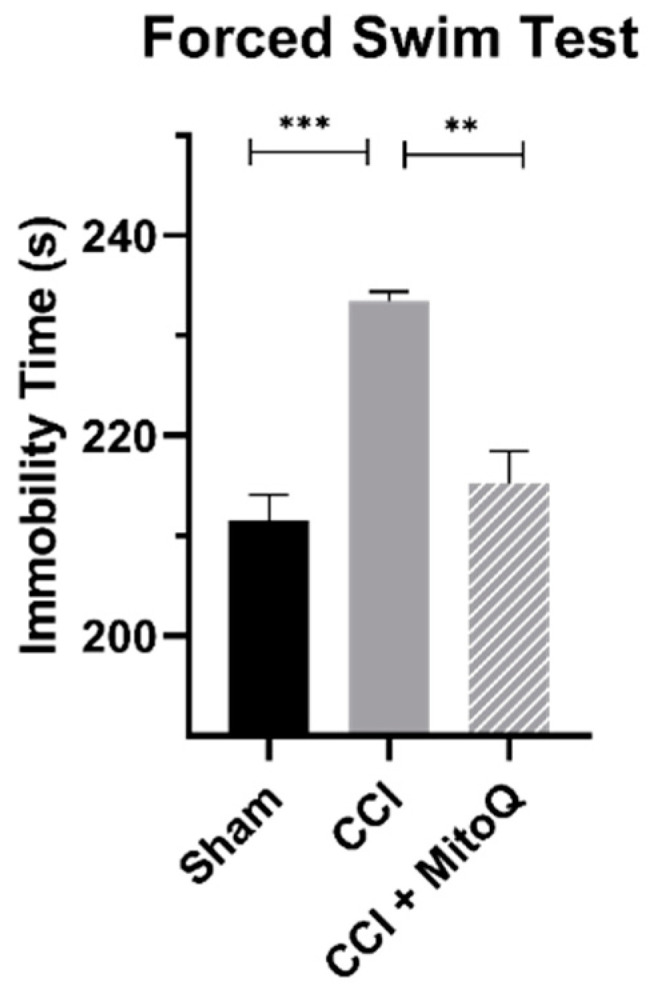
Depressive-like behavior is reduced by MitoQ. Animals were placed in a water container during the forced swim test to measure the immobility time as a sign of hopelessness. The CCI + MitoQ group performed better than the CCI group, with a decreased immobility time. Data represent the mean ± SEM (*n* = 10 per group). ** (*p* < 0.01), *** (*p* < 0.001).

**Figure 5 biomedicines-10-00250-f005:**
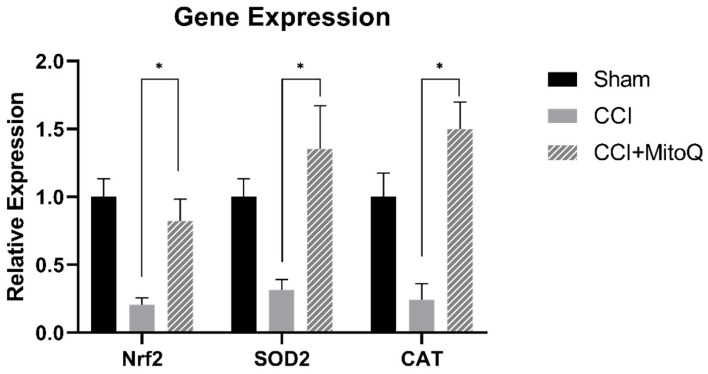
MitoQ improves expression of Nrf2, SOD2, and CAT in the cortex post-CCI. The mRNA levels of Nrf2, CAT, and SOD2 were determined using RT-qPCR in Sham (*n* = 4), CCI (*n* = 4), and CCI + MitoQ (*n* = 4). Data were normalized to β-actin. * (*p* < 0.05).

**Figure 6 biomedicines-10-00250-f006:**
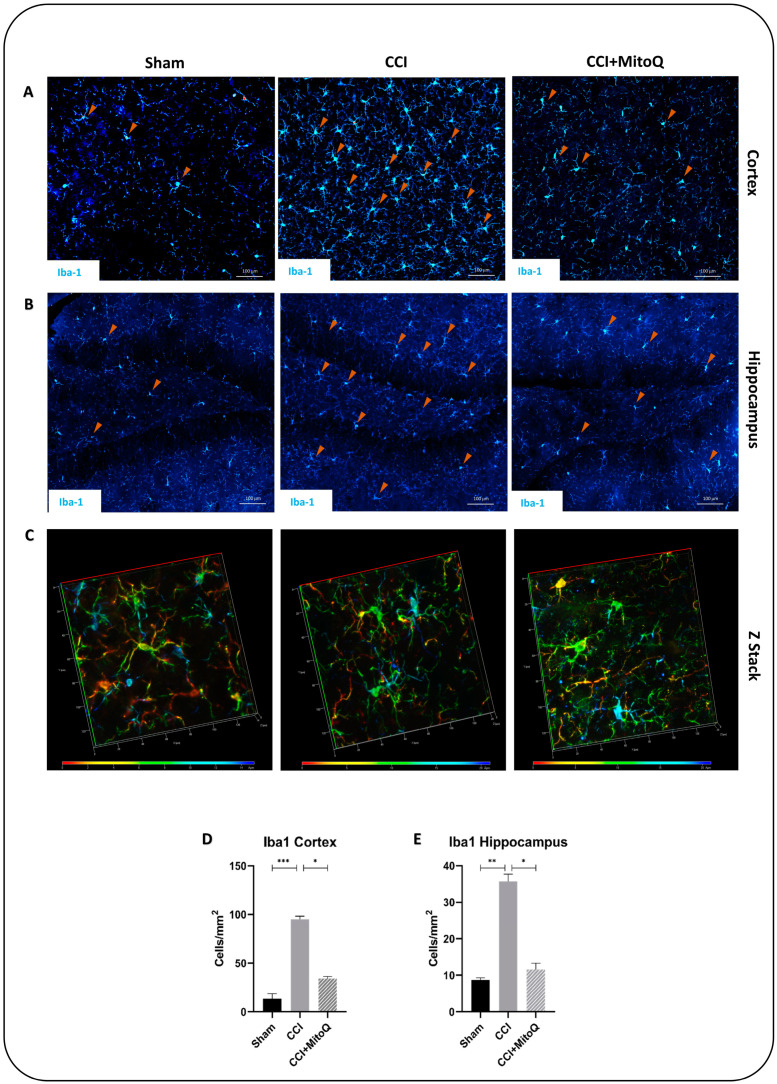
MitoQ decreases microglial activation in the cortex and the hippocampus. Microgliosis was assessed by immunofluorescence staining for Iba-1 (blue/green). (**A**) shows Iba-1-positive cells (arrowheads) across the three experimental groups in the ipsilateral cortex. (**B**) shows the same as (**A**), but in the ipsilateral hippocampus. Scale bar = 100 μm. (**C**) exhibits a depth-coded Z-stack of individual microglial cells to showcase different morphologies across the three groups. (**D**,**E**) are bar graphs showing quantification of Iba-1-positive cells per mm^2^. Iba-1-positive cells were counted, and then the obtained number was divided by the area of the selected field in an average of three fields per animal, four animals per group. * (*p* < 0.05); ** (*p* < 0.01); *** (*p* < 0.001).

**Figure 7 biomedicines-10-00250-f007:**
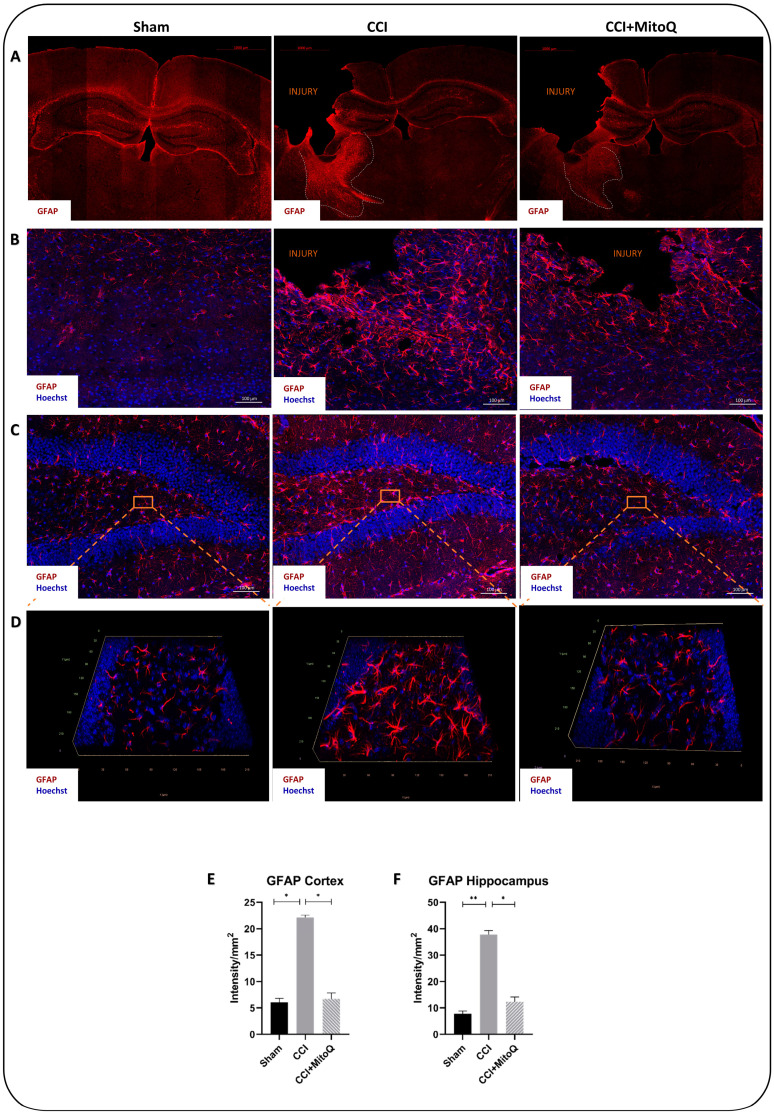
MitoQ ameliorates astrocytosis in the cortex and hippocampus post-CCI. Astrocytosis was assessed by immunofluorescent staining of GFAP (red). (**A**) shows the assessment of GFAP positivity in the entire brain section of all three groups. (**B**) shows GFAP-positive cells in the cortex and (**C**) in the hippocampus. Hoechst was used as a nuclear counterstain. (**C**) represents Z-stacks of sections in the hippocampal DG from all three groups to show different astrocytic morphologies. Scale bar = 100 μm. (**D**) shows a Z-stack of GFAP-positive cells in the hippocampus across the three groups. (**E**,**F**) are bar graphs showing the quantification of GFAP intensity per mm^2^. GFAP mean fluorescence intensities (MFI) of 3 fields per animal and 4 animals per condition were measured using the NIH ImageJ 1.41 software. Bar graphs display averages ± SEMs of GFAP MFI/mm^2^. * (*p* < 0.05); ** (*p* < 0.01).

**Figure 8 biomedicines-10-00250-f008:**
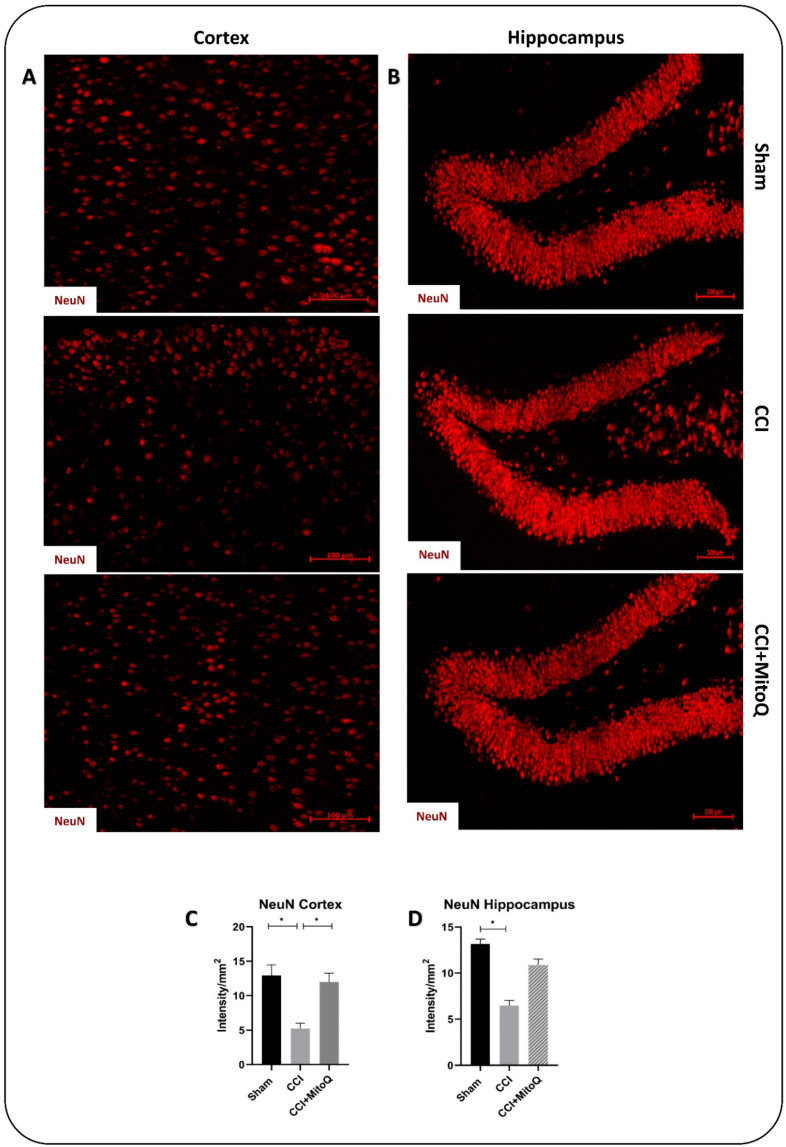
MitoQ decreases neuronal cell loss in the cortex. NeuN staining (red) was performed to assess the number of mature neurons. (**A**) shows representative fluorescent images of NeuN staining in the cortex from the different study groups. NeuN-positive cells were counted, and then the number was divided by the area of the field. (**B**) shows the same as (**A**), but in the hippocampus. GFAP mean fluorescence intensities (MFIs) of 3 fields per animal and 4 animals per condition were measured using the NIH ImageJ software. Scale bar = 100 μm. (**C**,**D**) are bar graphs displaying average MFIs ± SEMs of NeuN per mm^2^. * (*p* < 0.05).

**Figure 9 biomedicines-10-00250-f009:**
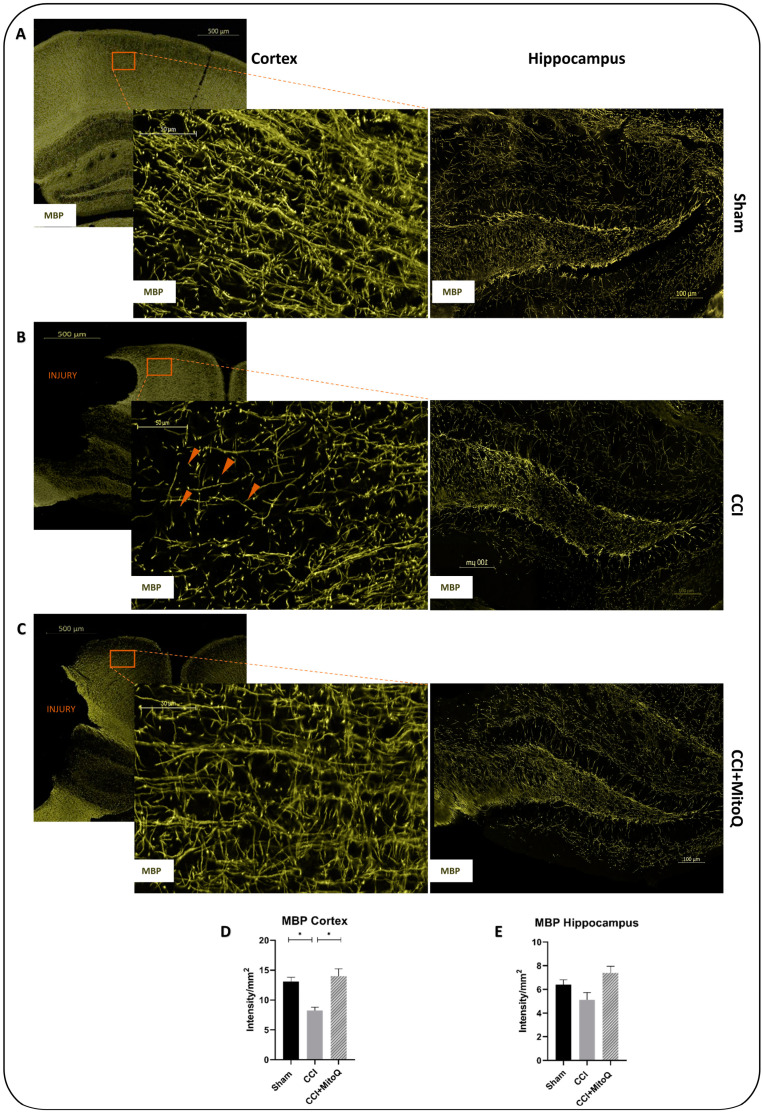
MitoQ decreases axonal shearing in the cortex after CCI. Representative fluorescent images of MBP staining (green) that were used to assess axonal morphology in the cortex and hippocampus for Sham, CCI, and CCI + MitoQ in (**A**–**C**), respectively. Major axonal blebbing and shearing are highlighted with arrowheads. Scale bars = 500, 100, 50 μm. (**D**,**E**) are bar graphs showing the quantification of the MBP intensity per mm^2^ in the cortex and hippocampus. MBP mean fluorescence intensities (MFIs) of 3 fields per animal and 4 animals per condition were measured using the NIH ImageJ software. (**D**,**E**) are bar graphs displaying averages ± SEMs of MBP MFI/mm^2^. * (*p* < 0.05).

**Figure 10 biomedicines-10-00250-f010:**
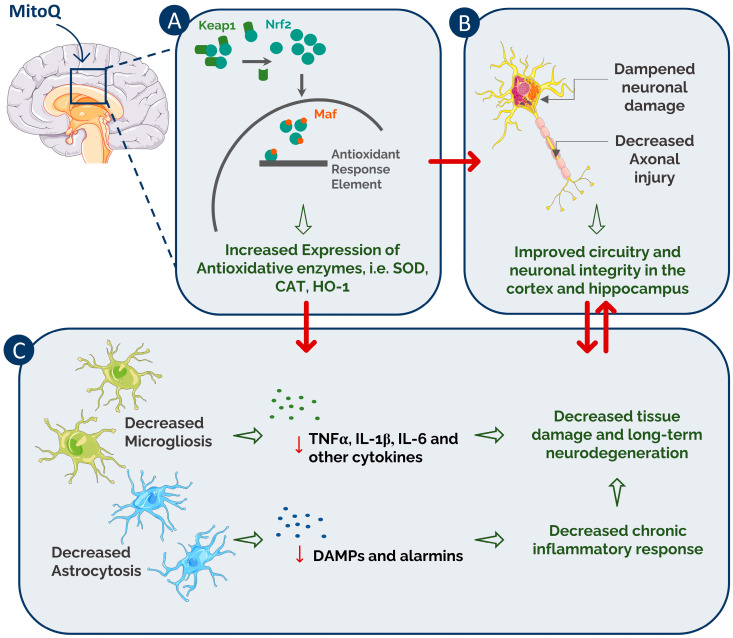
Summary of proposed MitoQ neuroprotective activity post-CCI. When MitoQ is administered, it exerts its effects via several pathways that eventually translate into improved neurological and cognitive functions. (**A**) MitoQ helps in increasing Nr2 expression and translocation into the nucleus by affecting its sequestration by the Kelch-like ECH-associated protein 1 (Keap1) protein. This allows Nrf2 to bind Maf proteins and interact with its antioxidant response element (ARE), leading to increased expression of antioxidative enzymes such as SOD2 and CAT. (**B**) Neurons are prone to increased apoptosis and axonal shearing due to the hostile environment created by the primary and secondary injuries. MitoQ helps decrease neuronal loss and maintains axonal integrity, resulting in improved circuitry in the brain. These effects translate into enhanced cognitive and neurological functions. (**C**) MitoQ helps decrease the activation of both microglia and astrocytes and therefore might help in decreasing chemical signals produced by both cells such as tumor necrosis factor α (TBFα), interleukin 1β (IL-1β), interleukin 6 (IL-6), cytokines, damage-associated molecular proteins (DAMPs), and alarmins. During a chronic inflammatory state, these would lead to damage to brain tissue and neurodegeneration.

**Table 1 biomedicines-10-00250-t001:** Evaluation criteria for Garcia neuroscore.

Criteria	Evaluation
Spontaneous Activity	Ability to approach all four walls of the cage
Limb Symmetry	Limb symmetry when held by the tail
Forepaw Outstretching	Outstretching symmetry of both forelimbs while the hindlimbs are kept in the air
Climbing	Ability to climb into and hang onto the cage
Body Perception	Reaction to stimulus while the mouse is touched on each side of the body with a stick
Vibrissae Touch	Reaction to stimulus while whiskers of the mouse are touched with a stick without entering the visual field

**Table 2 biomedicines-10-00250-t002:** Different primer sequences used in RT-qPCR.

Gene	Forward Primer (5′–3′)	Reverse Primer (5′–3′)
mSOD2	GGCCAAGGGAGATGTTACAA	GAACCTTGGACTCCCACA
mCAT	TGAGAAGCCTAAGAACGCAATTC	CCCTTCGCAGCCATGTG
mNrf2	CGAGATATACGCAGGAGAGGTAAGA	GCTCGACAATGTTCTCCAGCTT
mβ-actin	CAGCTGAGAGGGAAATCGTG	CGTTGCCAATAGTGATGACC

## Data Availability

The datasets used and/or analyzed during the current study are available from the corresponding author on reasonable request.

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
