# Peer review of "Mitoquinone Helps Combat the Neurological, Cognitive, and Molecular Consequences of Open Head Traumatic Brain Injury at Chronic Time Point"

_biomedicines, 2022, doi:10.3390/biomedicines10020250_

Round 1
Reviewer 1 Report
Over the past 20 years, there has been a significant increase in interest in the molecule Mitoquinone, the mechanism of protective action of which has not yet been fully studied and requires further research. The authors of the article experimentally studied the MitoQ cytoprotective effect on a model of brain injury. In the available literature, there are two works on the use of the molecule MitoQ in TBI, one of which is a literature review with the participation of the author of this study. During the discussion, the authors point out the strengths and weaknesses of their research, as well as the ways to solve them.
However, there are inaccuracies and flaws in the text that require correction.
in line 100, change the phrase “given every 3 days for a period of 30 days (until sacrificed)” on the phrase “three times per week 132 over 30 days, starting at 30 minutes post-injury”
in line 229, chapter 2.12. Immunofluorescence Staining 229. There is no description of the microscopy technique and there is also no description of the technique of quantitative evaluation of immunocytochemical studies, what were the conditions of euthanasia, which areas of the cerebral cortex and hippocampus are represented on the slices and taken into data processing. Due to the lack of precise indication of the field of immunocytochemical research, it is incorrect to discuss the results of the study.
All figures with immunostaining show very poor photo quality. Authors need to improve the quality of your slide presentation
in line 370-371, it is better to formulate the phrase like this: “Glial Fibrillary Associated Protein (GFAP) is an intermediate filament protein used as a marker of astrocytes. The intensity of its expression was used as a marker of active astrocytes”.
Reviewer 2 Report
The method need to be more detail. What extent skull was removed? Was dura opened? How much size of cortical impact?
Conclusion should be presented in main text and abstract.
